



# Influence of snow water equivalent on droughts and their prediction in the USA

Daniel Abel[1], Felix Pollinger[1], and Heiko Paeth[1]

[1]Institute of Geography and Geology, University of Wuerzburg, Am Hubland, 97074 Wuerzburg, Germany

**Correspondence:** Daniel Abel (daniel.abel@uni-wuerzburg.de)

**Abstract.** Droughts can result in enormous impacts for environment, societies, and economy. In arid or semiarid regions with bordering high mountains, snow is the major source of water supply due to its role as natural water storage. The goal of this study is to examine the influence of snow water equivalent (SWE) on droughts in the United States and find large-scale climatic predictors for SWE and drought. For this, a Maximum Covariance Analysis (MCA), also known as Singular
Value Decomposition, is performed with snow data from the ERA–Interim reanalysis and the self-calibrating Palmer Drought Severity Index (sc–PDSI) as drought index. Furthermore, the relationship of resulting principal components and original data with atmospheric patterns is investigated.

The leading mode shows the spatial connection between SWE and drought via downstream water/moisture transport. Especially the Rocky Mountains in Colorado (CR) play a key role for the central and western South, but the Sierra Nevada and
even the Appalachian Mountains are relevant, too. The temperature and precipitation based sc–PDSI is able to capture this link because increased soil moisture results in higher evapotranspiration with lower sensible heat and vice versa. A time shifted MCA indicates a prediction skill for drought conditions in spring and early summer for the downstream regions of CR on the basis of SWE in March. Furthermore, the phase of the El Niño–Southern Oscillation is a good predictor for drought in the southern US and SWE around Colorado. The influence of the North Atlantic Oscillation and Pacific North American Pattern
is not that clear.

## 1    Introduction

The aim of this study is to assess the influence of snow water equivalent (SWE) on droughts and find predictors for these processes in the United States of America (US). Droughts are extreme events causing high amounts of damage to humans (e.g. food and water scarcity, diseases), societies (e.g. unemployment, poverty), environment (e.g. damage for flora and fauna, wild
fires, low water and air quality), and economy (e.g. agriculture, forestry, industry) (Wilhite and Glantz, 1985; Wilhite et al., 2007), especially in arid and semiarid climates (Godfray et al., 2010). These impacts can occur on different geographical scales from local to regional via physical and economical connections (Ding et al., 2011). Ding et al. (2011) show that the damage of single drought events can reach costs greater than US$ 100 million very quickly with no upper limit.

If arid and semiarid climates have neighboring high mountains their water supply is dominated by snowmelt due to snow's
ability to act as a natural storage of wintertime snowfall in higher latitudes and/or elevations (Barnett et al., 2005). In the





Rocky Mountains, approximately two-thirds of annual precipitation are snowfall with maximum values of 85 % in some regions (Serreze et al., 1999). In spring and early summer, when ecosystems need the stored water most, it is set free by melt (Barry and Gan, 2011). This process becomes even more important with recent and future warming trends which result in less snowfall (Knowles et al., 2006), SWE (Gan et al., 2013; Dudley et al., 2017) and earlier melt with less runoff in the spring-summer phase (Stewart et al., 2005; Miller and Piechota, 2011) in which 50 to 80 % of annual runoff occurs (Stewart et al., 2004). These effects of global warming appear mainly in middle elevations because temperatures in higher ones remain below the dew point (Stewart, 2009). Not only the following season is affected by less available water but the groundwater storages are reduced, too, which is a substantial long-time effect (Dettinger et al., 2004)

Beside this hydrological aspect of snow, it affects the earth's energy balance due to the snow–albedo feedback. The albedo of snow shows a wide range between 25 and 95 %, depending on its temperature and age (Barry and Gan, 2011). Furthermore, snow has a higher heat capacity compared to bare ground, thus, the melting consumes energy which cannot heat the surface and overlying air (Sabade et al., 2011). Hence, snow influences circulation patterns due to its influence on the energy budget (e.g. Lo and Clark (2002)). Vice versa, circulation patterns influence regional temperature and precipitation, too. Some relevant atmospheric patterns for the US are the North Atlantic Oscillation (NAO), Pacific North American Pattern (PNA), and El Niño–Southern Oscillation (ENSO). The NAO acts mainly in the northeastern US where a positive (negative) phase corresponds with higher (lower) moisture transport and increased (decreased) precipitation and temperature (Hurrell, 1995). Furthermore, the NAO is part of the Northern Annular Mode which acts in the northwestern US (Thompson and Wallace, 2001). With a positive PNA, the temperatures in western North America (southeastern US) are above (below) normal (Leathers et al., 1991; Baxter and Nigam, 2013). During winter, less (more) precipitation occurs in the contiguous US (Leathers et al., 1991). A positive (negative) ENSO phase, called El Niño (La Niña), leads to wetter (drier) and cooler (warmer) conditions during winter with more (less) snow in the southern US, whereas the northeastern and northwestern US show warming (cooling) with less (more) snow (Seager et al., 2005). Furthermore, there is an overlapping effect of a negative NAO and El Niño with cold and snowy conditions in northeastern, central, and southern US due to shifted storm tracks and lower temperatures (Seager et al., 2010).

Because snow is part of surface hydrology this study deals with the hydrological drought type. As precipitation deficits need time to show up in parts of the hydrological cycle (soil moisture, streamflow, groundwater etc.) this type typically occurs out of phase with other drought types like meteorological, agricultural, or socio-economic drought (Karamouz et al., 2013). It can persist over very different time scales (Wilhite and Glantz, 1985). In general, hydrological droughts can be associated with a warm snow season with less SWE as a consequence of temperatures above 0 °C (Van Loon and Van Lanen, 2012). Accordingly, recent droughts in the study area were attributed to low amounts of winterly snow accumulation, e.g. in California (AghaKouchak et al., 2014; Griffin and Anchukaitis, 2014) and the Great Plains (PaiMazumder and Done, 2016). Note that these regions are expected to undergo further drying in the future (Ficklin et al., 2015).

Recent droughts, drying trends, and less SWE as water storage highlight the relevance of further research on the snow–drought relationship and drought predictability. Hence, the following questions are of high importance for the region's water supply: Is it possible to identify patterns of snow influence on drought? What are the main reasons of the relationship? Based on this, is drought prediction possible? Can atmospheric patterns act as predictors?



## 2 Data and method

### 2.1 Snow water equivalent data

The measurement of snow is done in situ or via remote sensing. In situ measurement guarantees a high accuracy but is selective and suffers from low spatial coverage (Barry and Gan, 2011), especially in high or remote mountain regions (Gillan et al.,
2010). Remote sensing is conducted optical or via passive microwaves. Optical data is of high to very high spatial resolution but supplies only snow cover as a binary variable (Barry and Gan, 2011). The latter method captures more aspects of snow but only on resolutions of 12 to 25 km (Chang et al., 1987; Dietz et al., 2012). Thus, we rely on a reanalysis dataset for this study which includes both measurements of snow.

The used SWE dataset — which represents the amount of water in mm if the accumulated snow melts instantaneously — is
from the ERA–Interim (ERA–I) reanalysis (1979–2015, monthly, 0.75°) (Berrisford et al., 2011; Dee et al., 2011). Thus, the data is coherent and complete in space and time which is a great benefit against relying solely on observations. Additionally, the regional to continental research scale supports this choice as the spatial resolution is appropriate. The ERA–Interim assimilation includes between 500 and 2000 daily snow measurements depending on season. Since July 2003 the daily snow cover product of NOAA/NESDIS Interactive Multisensor Snow and Ice Mapping System (IMS) with a resolution of $24\,km^2$ is used, too.
The assimilation generates a $10\,cm$ snow depth where the Integrated Forecast Model (IFS) — the base of ERA–Interim — is snow–free but the IMS–product is not. If the case is vice versa a snow depth of $0\,cm$ is assumed (Dee et al., 2011). As ERA–Interim provides only snow depth and snow density, SWE is calculated from these variables. Brun et al. (2013) found snowfall in ERA–Interim to be very reliable. Hence, it is an appropriate dataset to be used for snow modeling and statistical analysis.

Only the amount of SWE at the beginning of the melting period is relevant for drought as snow accumulates until then (Cayan, 1996; Clow et al., 2012). Mostly, April 1st is taken into consideration. However, due to the monthly data and a general underestimation of maximum SWE in the southwestern US at this date (Margulis et al., 2016) combined with maximum SWE amount by the end of March (Sobolowski and Frei, 2007) we consider the average SWE in March. We only take grid points with snow in at least 50 % of the years during the available data period into account to exclude extreme years.

### 2.2 Drought indices

Two drought indices (DIs) are used in this study to compare the results, but the focus lies on the first introduced index. The most important requirement is that the indices are multiscalar, thus, at least use temperature and precipitation to capture snow and its melt in an adequate way.

Firstly, the self–calibrating Palmer Drought Severity Index (sc–PDSI) (1950–2014, monthly, 2.5°) (Wells et al., 2004; Dai
and National Center for Atmospheric Research Staff, 2017), a revised form of the PDSI (Palmer, 1965; Alley, 1984), was used. The main difference between the revised and original index is that spatial comparability is not given in the original PDSI (Guttman, 1998) but in sc–PDSI (Wells et al., 2004). Furthermore, the duration factors are revised to a higher weighting of the previous PDSI and a lower of the actual moisture anomaly, the Z–value. Vicente-Serrano et al. (2010b) found that the actual





sc–PDSI value shows a memory of 9–18 months in North America. Both PDSI versions are based on monthly temperature and precipitation and use the Thornthwaite scheme (Thornthwaite, 1948) for potential evapotranspiration (PET).

Secondly, the Standardized Precipitation Evapotranspiration Index (SPEI) (Vicente-Serrano et al., 2010a), version 2.5 (1901–2015, monthly, 0.5°) (Beguería et al., 2014; Beguería and Vicente-Serrano, 2017), a revised form of the Standardized Precip-
itation Index (SPI) (McKee et al., 1993), is considered. The SPEI can be investigated on scales of 1–48 months which is a strong advantage over PDSI and sc–PDSI with their constant duration factors as different lengths of droughts can be investigated (Guttman, 1998; Wells et al., 2004). Due to standardization a spatio-temporal comparison of values is possible (Vicente-Serrano et al., 2010a). Since version 2, the PET is calculated with the Penman–Monteith scheme (Monteith, 1965). This performs better but requires more variables, namely net radiation, moisture, and wind speed in the boundary layer (Be-
guería et al., 2014). We employ the SPEI of 6 months (SPEI06) to investigate shorter periods than sc–PDSI and compare the results. In addition, the analysis were made with a SPEI of 1 month (SPEI01) to exclude the memory of the index.

Low (High) index values of the indices indicate drier (wetter) conditions. All indices contain monthly values averaged over the hydrological year (October–September) to close the hydrological cycle. Solely for the predictive potential (section 3.4) the non-averaged monthly values were used.

## 2.3 Atmospheric indices

The temporal variation of the atmospheric patterns influencing the study area is represented by atmospheric indices (AIs). We consider the North Atlantic Oscillation Index (NAOI (Jones, 1997; CRU, 2017)), the Pacific North American Index (PNAI (Leathers et al., 1991; NOAA / National Weather Service, 2017)), and the Multivariate ENSO Index (MEI (Wolter and Timlin, 1993, 1998; ESRL, 2015)). The first index uses the standardized pressure difference between Gibraltar and southwestern
Iceland whereas a positive value represents a positive NAO anomaly (Jones, 1997). The PNAI is calculated with the difference of standardized pressure anomalies in 700 mb height between the sum of anomalies at two northern locations in Montana and the North Pacific and the eastern Gulf of Mexiko (Leathers et al., 1991). The MEI is expressed by the first principal component (PC) of six standardized variables. Positive (Negative) values mark El Niño (La Niña) (Wolter and Timlin, 1993).

## 2.4 Maximum Covariance Analysis

The Maximum Covariance Analysis (MCA), also called Singular Value Decomposition, is performed to identify the spatio-temporal fields or modes of two variables (here SWE and DI) with coupled variabilities. In the following, there is a short description of the method. For further details, see Bretherton et al. (1992), Wallace et al. (1992), and Björnsson and Venegas (1997).

MCA is based on two matrices $S(n \times p)$ and $P(n \times q)$ with $n$ time steps and $p$ and $q$ grid points (locations). Thus, the
method can deal with different spatial data resolutions. We consider time series of standardized anomalies. Subsequently, the joint covariance or correlation matrix is calculated. Correlation matrix is advantageous, if one variable varies substantially




stronger than the other due to its effect on the covariances. The resulting matrix $C(p \times q)$ gives the definition of MCA:

$$C = ULV^t. \tag{1}$$

$U(p \times p)$ and $V(q \times q)$ are called the left and right patterns and contain the singular vectors of the matrices $S$ and $P$ in their columns. $L(p \times q)$ is a diagonal matrix containing the singular values.

On the basis of this, the PCs, often called the expansion coefficients, are calculated by projecting $S$ and $P$ onto the singular vectors. Strong correlations between the PCs of the left and right patterns represent a coupling between the variables (Cherry, 1997). To calculate the percentage of explained covariance, the squared covariance fraction (SCF), of the $i^{th}$ PC, one divides the $i^{th}$ squared singular value $l_i$ by the sum of all squared singular values.

To interpret the spatial connections, homogeneous and heterogeneous correlation maps are produced. The first type is the vector of correlations between the data of $S$ and $P$, respectively, and the $i^{th}$ PC of the *same* dataset and shows the locations of active covariance patterns of the variables (Wallace et al., 1992). The latter ones show the vector of correlations of $S$ and $P$, respectively, and the $i^{th}$ PC of the *other* dataset. They represent the spatio-temporal correlation between the first variable (here SWE) and the second one (here DI). Hence, they show how well the second field can be predicted by the PC of the first field. Generally, the homogeneous correlations are stronger than the heterogeneous ones (Bretherton et al., 1992; Wallace et al., 1992).

The results of MCA should be treated with caution as the patterns in the correlation maps might be solely a mathematical construct or the underlying system is not completely understood. Thus, interpretation has to be supported by a (geo)physical explanation (Newman and Sardeshmukh, 1995; Cherry, 1997). As MCA maximizes the covariance between multivariate variables, it is possible that the resulting homogeneous and heterogeneous patterns are closely correlated but do not represent major modes of the variability of each original variable (Wallace et al., 1992; Cherry, 1996). To encounter this, a PCA was performed to compare the PCs and patterns and confirm or contradict the results of the MCA (Cherry, 1997). The PCA reduces the data dimensionality by finding linear combinations that maximize the variance. It acts on one field instead of two which leads to an independent view on the variances of each variable (Wilks, 2011).

## 3 Results

First, regression analysis (not shown) was performed to evaluate, if the datasets show the observed and described trends of recent literature. Due to high variability during the considered period, which might overlap the trends, linear trends were mostly not significant at the 5 % level. Nevertheless, the resulting patterns for SWE and DIs resemble those from the literature with declining SWE in southwestern and mountainous US and drying conditions in the south, southwest, and central US. Thus, the data capture the recent developments.





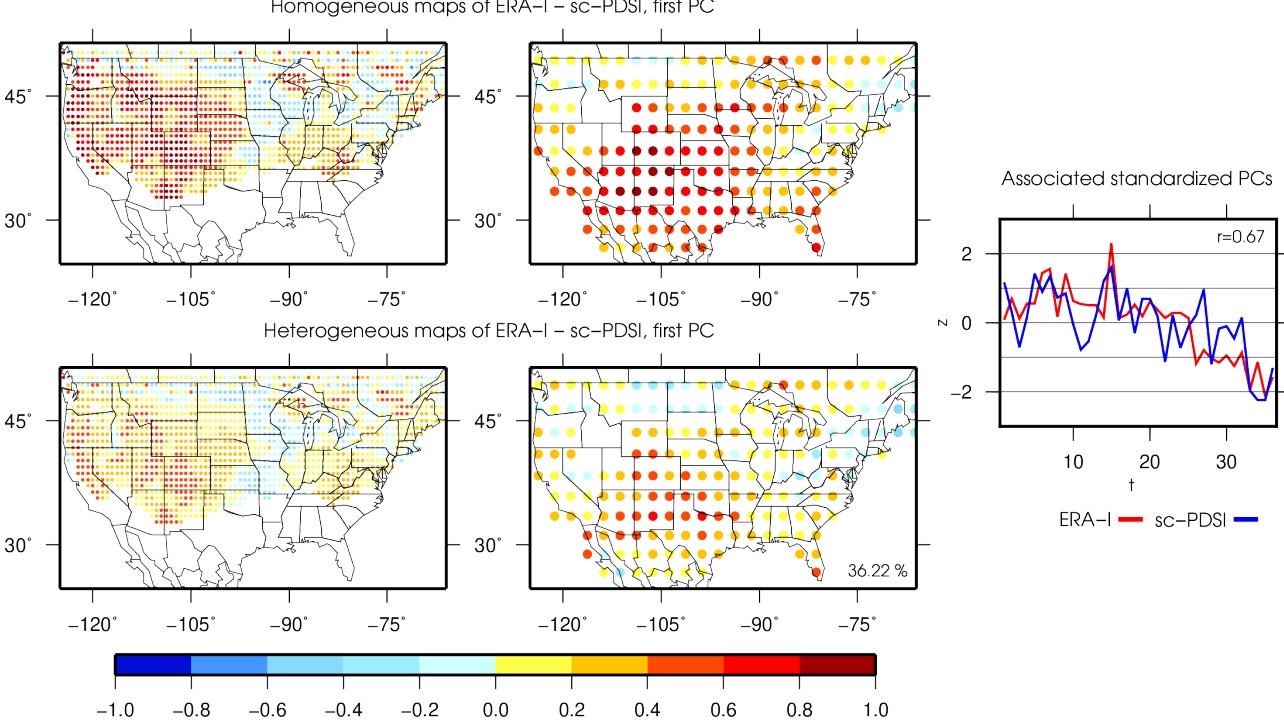

**Figure 1.** First homogeneous (top) and heterogeneous (bottom) correlation maps of ERA–I (left) and sc–PDSI (right) with associated PCs. Additionally, the correlation between the PCs is given in the xy-plot. Explained covariance is given in the heterogeneous sc–PDSI map.

## 3.1 Correlation maps of MCA

The correlation maps of the first PCs and the original data as well as the associated PCs are shown in figure 1. The PCs' trend confirm the general tendency of less SWE and drier conditions over the study period and area. Furthermore, they show a strong correlation of 0.67 which demonstrates a coupling of SWE and sc–PDSI. The PCs explain 36.22 % of the total covariance.

5      The homogeneous correlation maps (top row) illustrate the high variability of SWE west of 100° W in the Rocky Mountains and neighboring mountain ranges. High variances of sc–PDSI are located in the arid southwestern US and the Great Plains as well as in parts of Florida. These centers of variance fit well with the general occurrence of the variables and the patterns of the regression analysis. Thus, the patterns in the homogeneous correlation maps capture the dominating trends well.

     The heterogeneous maps (figure 1; bottom row) show the spatial influence of SWE on sc–PDSI. The patterns are weaker

10     but similar, thus, the cross-validation works. Where correlations of SWE and sc–PDSI in the heterogeneous maps are strong, the variables are associated with each other. Thus, SWE in regions with high correlations has an influence on sc–PDSI in highly correlated regions. High correlations of SWE mainly occur in the southwestern US, for sc–PDSI they are located in the southwestern and central US.

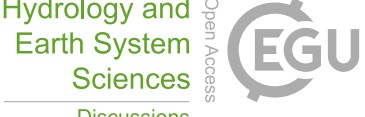

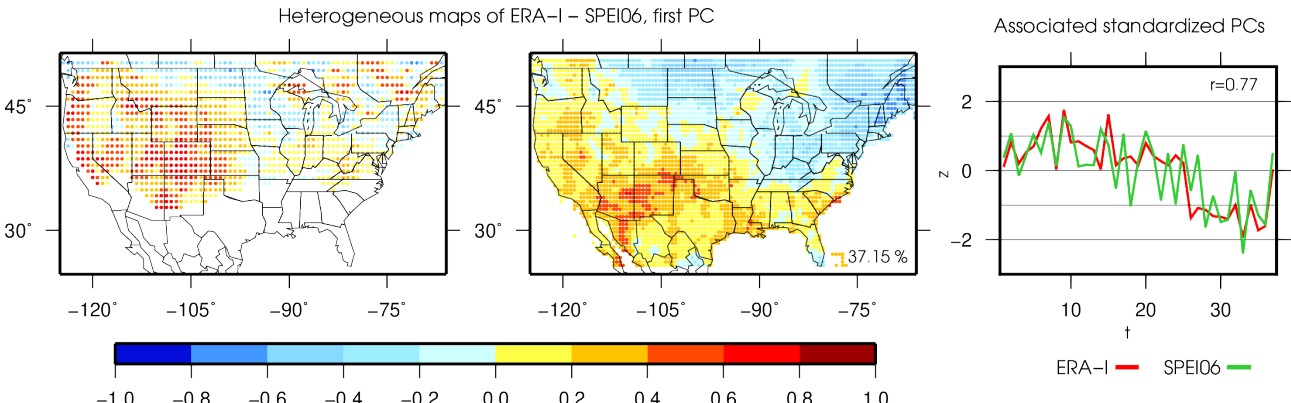

**Figure 2.** Same as figure 1 but solely with heterogeneous maps of ERA–I and SPEI06.

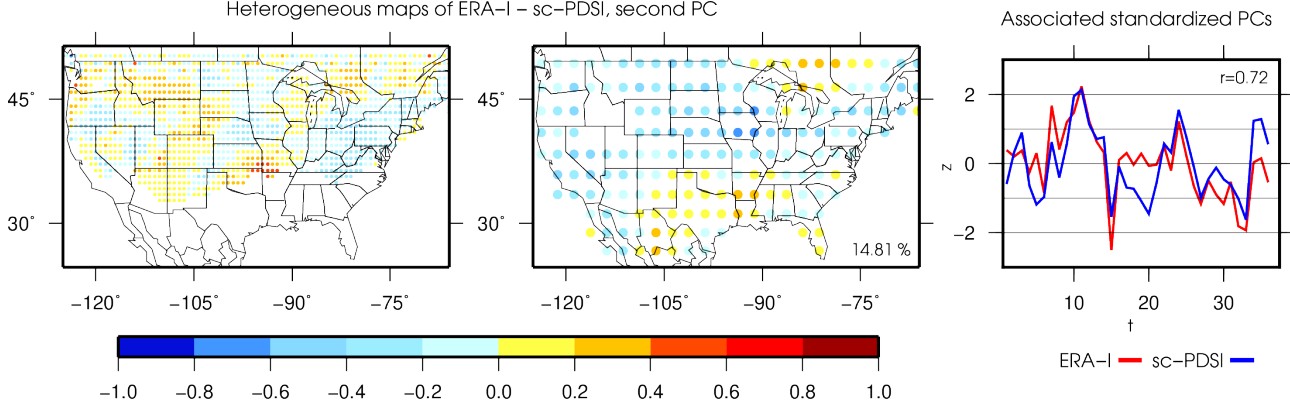

**Figure 3.** Same as figure 2 but for the second PC of ERA–I and sc–PDSI.

To validate the occurring patterns and investigate a shorter drought period of 6 months, the MCA was performed with SPEI06, too. Figure 2 shows the resulting heterogeneous maps of the first PC of SWE from ERA–Interim and SPEI06. The PCs show the same negative trend as in the first examined situation but a stronger correlation. Also, the SCF is marginally higher. Hence, both drought indices capture the same effects. Generally, the heterogeneous maps of SWE show similar patterns. Here,

5 the extent of medium to strong correlations is greater around the Rocky Mountains and southwestern regions. The patterns of SPEI06 are more heterogeneous and show a dipole structure with centers in the southwest and northeast. The southwestern center shows the same structure as in the sc–PDSI situation but of smaller extent. Consequently, the same observations as in the previous case can be made for the southwest and the southeast where SWE around the mountain areas influences SPEI06 in the southwestern and central US. Furthermore, there seems to be a negative influence between snow in the southwest and

10 SPEI06 in the northeast and central north but of lower extent.

Figure 3 shows the second PC of SWE with sc–PDSI and the resulting heterogeneous maps. The PCs still show a slightly decreasing trend as it was already observed in the leading mode. Furthermore, they also are highly correlated (0.72) which shows





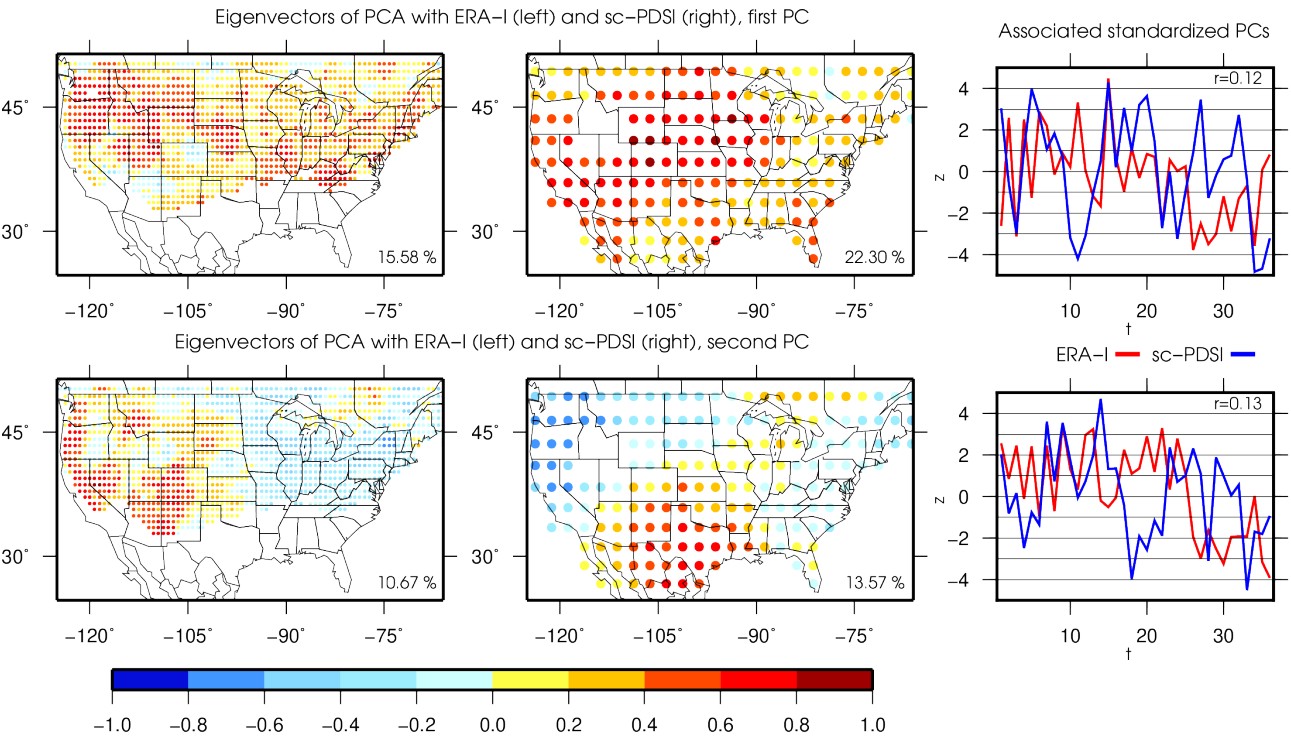

**Figure 4.** First (top) and second (bottom) eigenvectors of ERA–I (left) and sc–PDSI (right) with associated PCs. Explained variances are shown in the respective map, correlations in the xy-plot.

the variables' coupling. The second PC explains 14.81 % of covariance. Considering the heterogeneous correlation maps, the patterns are of remarkably weaker expression than previously. However, the patterns of SWE still show a comparable structure as in the first PC with centers of positive correlations in the mountain ranges and parts of the Great Plains. Weak negative correlations are located in the east around the Appalachian Mountains which differs from the first PC. The heterogeneous map

5    of sc–PDSI shows differing structures with a weak positive center in the central South but mainly negative correlations at the other locations. Medium to strong negative correlations exist between 37° and 45° N from the west coast to 90° W. Therefore, similar patterns of SWE in the first two modes show a different influence on sc–PDSI in the second PC which will be examined subsequently. The second PCs' performance of SPEI06 (not shown) lead to comparable results.

## 3.2   EOFs of PCA and comparison of PCs

10   The PCA was performed to address the mentioned limitations of MCA and verify the already described findings of MCA. The top row of figure 4 illustrates the first eigenvectors of SWE of ERA–Interim (left map) and sc–PDSI (right map) with the associated standardized PCs. The bottom row shows the same but for the second eigenvectors and PCs. The explained variances are located in the bottom right corner of each map. The PCs show a much higher variation than for MCA. However, there is a decreasing trend in the PCs of both variables and in both observed modes.





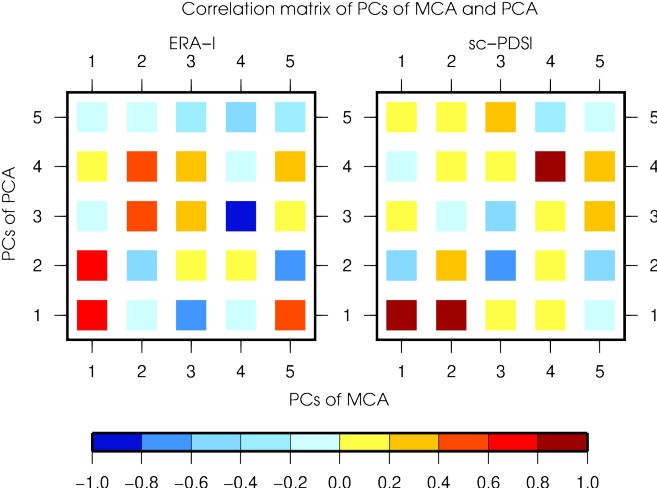

**Figure 5.** Correlation matrix of the five leading PCs of MCA (x-axis) and PCA (y-axis) of ERA–I (left) and sc–PDSI (right), respectively.

The first eigenvectors of ERA–Interim highlight the regions of the northern and central Rocky Mountains and Appalachian Mountains as well as parts of the Great Plains and the northeast. The second eigenvectors in the bottom row have a dipole structure with positive correlations in the Coastal Mountains and Rocky Mountains around Colorado (CR) and negative ones in the eastern part of the US. Comparing these patterns with the first heterogeneous correlation map of ERA–Interim in figure 1,

it can be seen that the patterns there are a composite of the first two eigenvectors.

Subsequently, the relation between SWE and sc–PDSI, respectively, and the two dominating patterns of the study area is investigated. Taking the first eigenvectors of sc–PDSI (figure 4) into account, there are mainly medium to strong positive correlations in the Great Plains and southern and southwestern US. The second eigenvectors are more heterogeneous with strong positive correlations in the central south which become weaker in the southern Great Plains. Strong negative ones are

located in the northwest with weaker expressions in the Rocky Mountains as well as at the east coast. Again, a comparison with the first heterogeneous map of sc–PDSI (figure 1) is done. The pattern there has lower amplitudes but the central and western south are highlighted by positive correlations, too. Furthermore, the second map of sc–PDSI in figure 3 shows a similar but much weaker structure as the first eigenvectors of PCA. There is some similarity between the second modes of PCA and MCA, too. The first eigenvectors of SPEI06 (not shown) and the sc–PDSI pattern look alike.

To confirm or oppose the structures' similarity of the two conducted eigenvalue techniques and modes, figure 5 shows a correlation matrix of the first five PCs of each technique. The left plot refers to the PCs of ERA–Interim and the right plot to those of sc–PDSI. PCs of MCA are given on the x-axis, PCs of PCA on the y-axis. Focusing on the left SWE-plot, the assumption that the leading PC of MCA shows a composite of the first and second PC of PCA can be confirmed due to high correlations between the time series. Furthermore, it can be seen that the second PC of MCA represents parts of the third and

fourth PC of PCA as well as inverse parts of the second PC — all with medium to high correlations. However, the explained variance of higher-order modes of PCA is less than 10 % and, thus, not further explored.





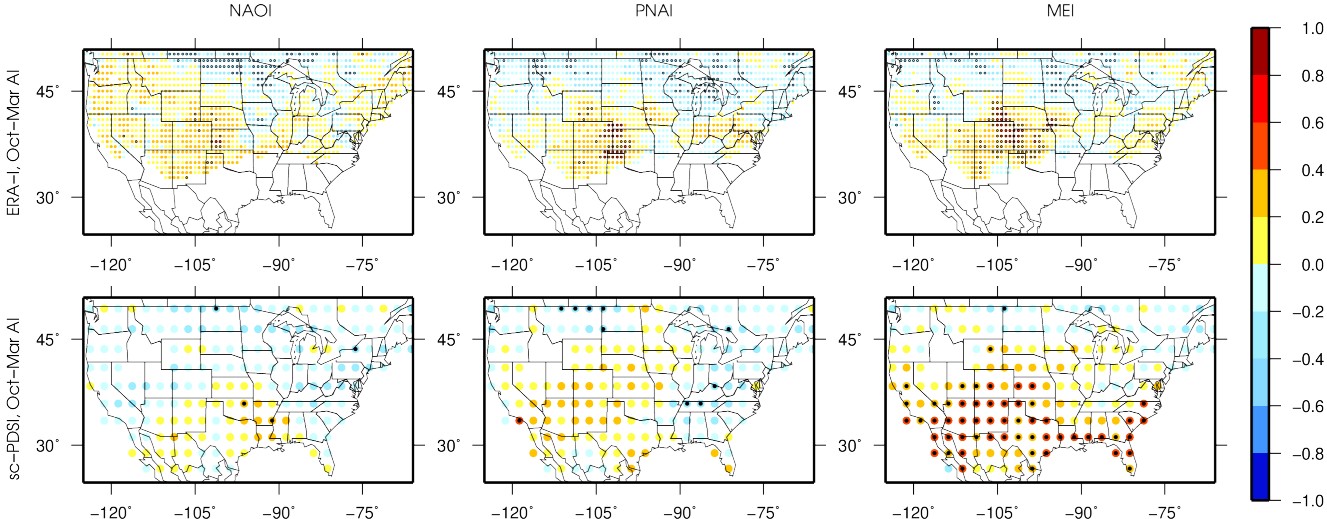

**Figure 6.** Correlation maps of ERA–I (top) and sc–PDSI (bottom), respectively, with averaged NAOI (left), PNAI (middle), and MEI (right) over Oct–Mar. Significant values at a significance level of 5 % are marked with black dots.

Taking a view on the right plot, it is obvious that the first and second mode of MCA are summarized in the first mode of PCA. The multiple correlation of this is found to be nearly 1. Thus, the previously made assumption can be confirmed, too. Additionally, the third mode of MCA shows a strong negative correlation with the second mode of PCA. From this, it can be concluded that for our data the MCA is generally able to capture the same patterns as PCA and, hence, represent not only strong correlations but also an important feature of each individual variable. Furthermore, it gives an additional scope of maximizing the correlations between the datasets. As the first two modes of MCA represent basically the same patterns as these of PCA, the further investigation is done on the basis of the PCs of MCA.

### 3.3 Influence of atmospheric patterns

Subsequently, the link between SWE and sc–PDSI, respectively, and three dominating atmospheric patterns in the study area is investigated. Figure 6 shows correlation patterns of AIs with SWE and sc–PDSI. As the atmospheric patterns mainly act during the winter half year the correlation maps are produced with the averaged indices of Oct–Mar. For SWE and NAOI, we find positive correlations in the western US, around the area of Colorado and east of that as well as in the northeastern US which indicates more snow during a positive phase. Negative ones are located west of the Great Lakes. However, only a small number of correlations is significant. For PNAI, stronger correlations exist around Colorado but north of 40° N negative ones dominate. The correlations between SWE data and MEI are quite similar to the previous ones with PNAI. Thus, there seems to be a medium influence on or of snow in the Colorado region but not in the high mountains or in the northwest which might be represented in the second PC.

Considering the sc–PDSI–NAOI relationship, solely the central South is marked by positive correlations whereby the rest shows negative ones. However, all are weak and not significant. For PNAI, the positive centers are located in the central south,



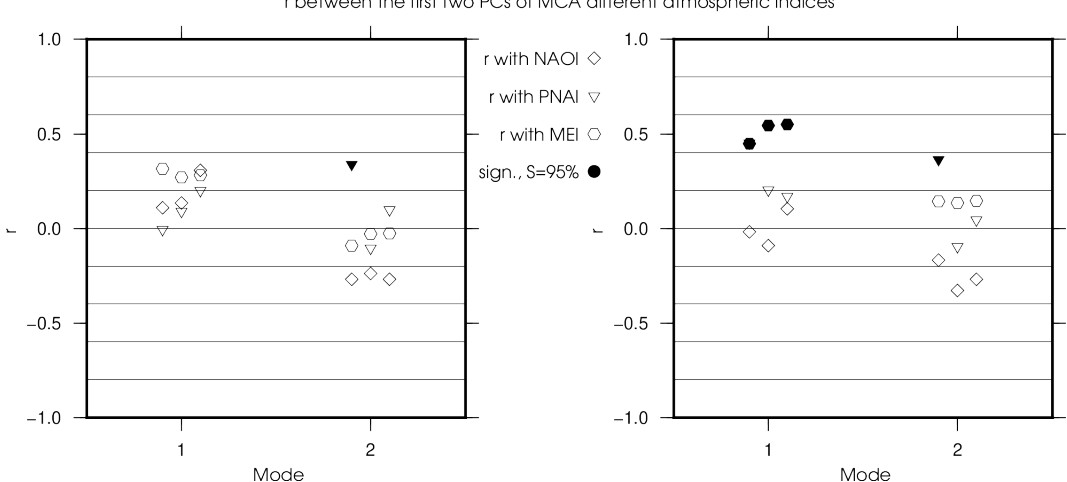

**Figure 7.** Correlation between the first two PCs of ERA–I (left) and sc–PDSI (right) with the indices of three dominating circulation patterns (North Atlantic Oscillation Index (NAOI), Pacific North American Index (PNAI), and Multivariate ENSO Index (MEI)). The indices are averaged over different periods: annual (shifted to the left), DJF (center), and Oct–Mar (shifted to the right). Significant correlations are plotted with filled symbols.

Mid West, and with strongest expression in the southwest. Additionally, a negative center is located in the southeast. The MEI pattern shows a clear influence of ENSO on sc–PDSI in the southern US which is also represented in the first PC.

Figure 7 shows the correlations between the first two PCs of MCA of ERA–Interim (left) and sc–PDSI (right) and the three AIs. The indices are averaged over the annual, DJF, and Oct–Mar periods. Significant correlations at a significance level of
5  5 % (exceeding an absolute value of 0.33 with 36 observations) are marked by filled symbols. The leading PC of SWE shows no significant correlations with the AIs while the second mode has a significant correlation with the annual PNAI. Considering sc–PDSI, there are significant correlations for the first mode and MEI at all observed periods. The second PCs' correlation is significant with annual PNAI again. However, with a little longer time series there might be more significant correlations in the first PC of SWE (MEI, NAOI) and second of sc–PDSI (NAOI) as they only fall marginally below the significance threshold.
10  Furthermore, it has to be taken into account that the PCs are time series which represent the contiguous US, where the high loadings in the homogeneous maps of figure 1 have the highest influence on their generation. The atmospheric patterns on the other hand mostly act on regional and not on continental scale. Thus, a more regional analysis might lead to higher correlations between the PCs and AIs.

## 3.4   Drought predictability

15  To investigate the influence of snow on droughts on monthly scales and a possible prediction approach, the MCA is performed in a time shifted way to predict the sc–PDSI, the SPEI06, and the SPEI01. The latter is calculated solely with values of the specific month, thus, there is no memory of the index inherent to the method. This is important as the DIs are no longer





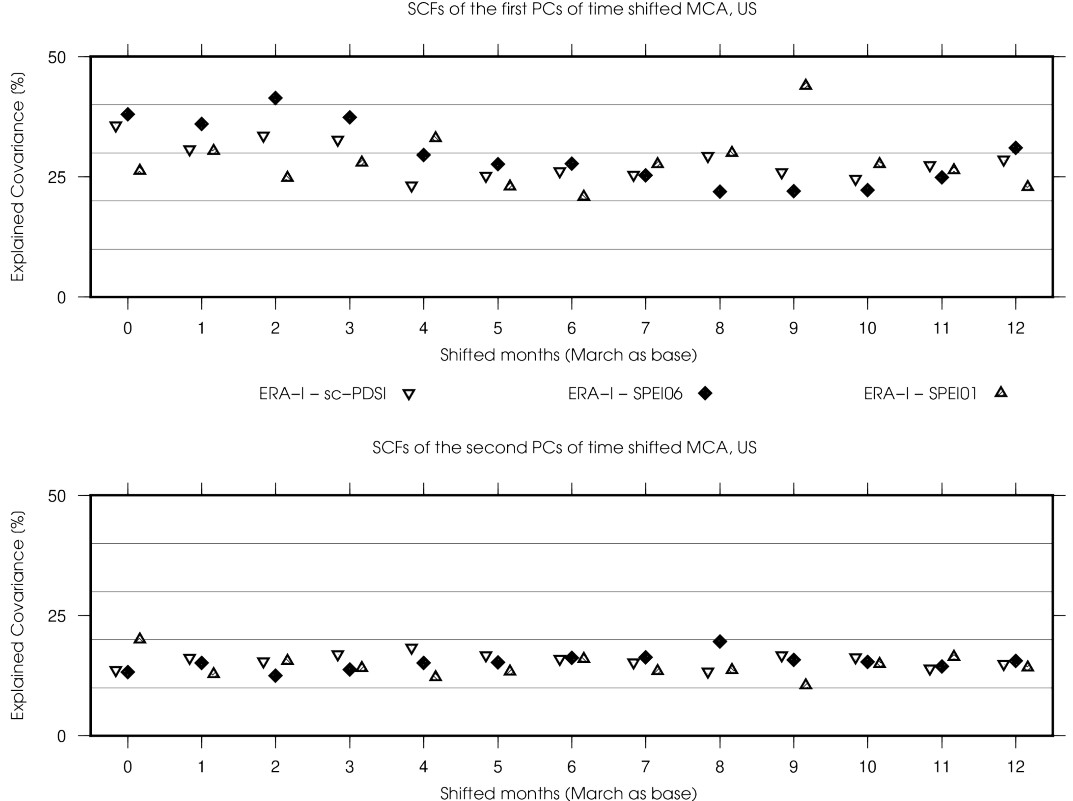

**Figure 8.** SCFs of the first two modes of time shifted MCA with SWE in March as base (month 0) and three monthly shifted DIs (sc–PDSI, SPEI06, SPEI01).

considered for the hydrological year but on a monthly time scale. Hence, the time shifted MCA is performed on the base of SWE in March (month 0) and the DIs of the same and 12 consecutive months. Figure 8 shows the SCFs of the first (top) and second (bottom) mode of this time shifted approach.

The SCFs of the leading mode show high and slightly increasing values in the first months and decreasing ones afterwards.
5   Focusing on single indices, sc–PDSI has a SCF between 30 and 40 % from March to June where the predictive potential is highest. For April, the skill is the lowest one for these four months. SPEI06 shows a similar development but with higher SCFs in the first months. This allows the same conclusion of a predictive skill from March to June with an extension to July. After July, the SCFs range constantly between 20 and 30 %. In terms of both indices, the memory has to be taken into account which is for sc–PDSI between 9–18 months and for SPEI06 at 6 months. The corresponding patterns (not shown) are of similar shape
10  as for the annual MCA which confirms the relationship between SWE and DIs in the mentioned regions. SPEI01, without any memory, shows slightly different characteristics. There, only for April and July the SCF of 30 % is exceeded in the first five months where the other DIs show their lowest values. After July it is constantly between 20 and 30 %. However, the value of December (month 9) is extraordinary as it reaches 44 % which is the highest SCF of all DIs.





The SCFs of the second modes show approximately constant values which vary only marginally. Nearly all values range between 10 and 20 % of explained covariance and a clear time step over which a prediction might be made does not exist. Nevertheless, SPEI01 shows the highest variation which is not surprising due to its definition over one month.

## 4 Discussion

### 4.1 Relation between SWE and drought and predictive potential

Considering the patterns of figure 1, one observes that SWE in the Colorado and Northern Rocky Mountains and Sierra Nevada influences sc–PDSI in the central and southwestern US. These areas mainly lie downstream of important rivers with nival regimes and source in CR, namely Colorado River to the southwest, Rio Grande to the south and Arkansas River to the southeast. Additionally, there is a weaker connection between the Appalachian Mountains and their southward drainage area. Thus, the connection of SWE and sc–PDSI may work via drainage of the mentioned systems. The one in the Northern Rocky Mountains might be not that clear due to generally wetter and cooler conditions. The influence of the Sierra Nevada on California and Nevada can not be explained with drainage at the spatial resolution of sc–PDSI but of SPEI06 in figure 2.

Hence, for the correlation patterns of the leading mode resulting from MCA, the hypothesis of the relation between SWE and DIs via melt and drainage is proposed. This thesis seems obvious in the first moment but the arising question is why this relation can be observed so clearly using the DIs based on temperature and precipitation data and is obviously not restricted to hydrological variables like runoff or soil moisture (SM)

Garcia and Tague (2015) have shown that subsurface water storage capacity (AWC) is an important driver for annual evapotranspiration (ET) in nival catchments of the western US. They found that a later (earlier) melt due to lower (higher) spring temperatures leads to an increased (decreased) recharge of subsurface water storages, e.g. soil moisture, and less (more) loss of stored water to runoff. Vice versa, temperature is also influenced by the prevailing amount of SWE. Following them, there will be a higher (lower) annual ET due to the available moisture – which is besides energy the second limiting factor of ET (Seneviratne et al., 2010; Garcia and Tague, 2015). Regarding the influence of SM on temperature (in particular the review of Seneviratne et al. (2010)), this leads to a decreased (increased) temperature as ET reduces (enhances) the sensible heat and acts vice versa with latent heat. The effect of spring SM on summer droughts or heat waves has already been found in the US (e.g. Durre et al. (2000); Zhang et al. (2009)) and also Europe (e.g. Diffenbaugh et al. (2007); Fischer et al. (2007); Jaeger and Seneviratne (2011)). We assume that SWE and its melt water is responsible for filling the subsurface storages in the downstream regions of the CR. However, Hamlet et al. (2007) showed that a missing recharge by snowmelt can be compensated by summer precipitation.

Due to this indirect effect of SM on ET the PET schemes of the DIs play a substantial role for the question, whether the indices are able to capture the above explained processes. The Thornthwaite scheme of sc–PDSI with its monthly mean temperature and latitude might be able to represent this as it considers temperature. The Penman–Monteith scheme of SPEI also considers moisture and, hence, might represent sensible as well as latent heat. Thus, it might be able for the DIs to capture the distribution of melt and resulting SM via its influence on ET and temperature. Additionally, the importance of snow on





downstream regions' water supply is already outlined for the CR and its big rivers (Serreze et al., 1999) as well as the Sierra Nevada (AghaKouchak et al., 2014). The influence of SWE in the Appalachian Mountains on droughts in downstream areas has not been investigated sufficiently, yet. Nevertheless, there are studies of affecting the Ohio River (Coleman and Rogers, 2003). Further, the mountains' snow water equivalent might have an influence on droughts for the southerly Atlantic-Gulf

Basin (Kelly et al., 2012) where they occur frequently (Maxwell and Soulé, 2009).

The performance of MCA with monthly DI data showed that there might be a predictive potential for droughts influenced by the amount of SWE for spring and early summer with explained covariances above 30 %. The regions of influence are captured well by the heterogeneous maps of figure 1 and 2.

### 4.2   Influence of atmospheric patterns

It was shown that especially MEI has a strong influence on drought in the southern US (Rajagopalan and Cook, 2000) and SWE around Colorado (Seager et al., 2010). For SWE and NAOI, there might be the mentioned overlapping effect with ENSO in the central US. Furthermore, the northwestern region shows higher amounts of SWE during a positive phase due to the Northern Annular Mode being partially expressed via NAOI (Thompson and Wallace, 2001). Considering PNAI, the positive phase is associated with a higher snowfall/precipitation ratio in southeastern US and less SWE in the central north (Sobolowski and

Frei, 2007) and northwest (Leathers et al., 1991; Abatzoglou, 2011) which fits well to the shown correlation patterns (figure 6). However, we expected a clearer relation between the two northern circulation patterns and SWE. This would result in better prediction potential for SWE in March and the amount of stored water. Nevertheless, MEI has the highest potential for forecasting drought conditions in the southern US.

### 4.3   Further investigations

Not dealt with in this paper, we also used the SNOw Data Assimilation System (SNODAS) (Barrett, 2003; NOHRSC, 2004) for SWE data in the US. We found comparable results, however, these are not shown due to its short data period since 2003. Indeed, the spatio-temporal resolution of 1 km and 1 h is outstanding and shows – in agreement with Clow et al. (2012) and Hedrick et al. (2015) — a great research potential.

Further, the methodology presented in this study was employed for Central Asia (not shown) as the requirements of nival

regimes (Unger-Shayesteh et al., 2015) and related droughts (Pritchard, 2017) are also given there. Nevertheless, the results of MCA were not that clear as for the US. This might be due to the significant amount of water which is not only provided by snow but glaciers, too. For the investigated variable SWE, the relationship between temperature and SWE and glaciers, respectively, is vice versa. Thus, with higher temperatures increased glacier melt might balance lower amounts of SWE (Braun and Hagg, 2009). Further research has to focus on this subject.





## 5   Conclusions

The study demonstrates how MCA can be used to find spatio-temporal links between SWE and drought in the contiguous US. Furthermore, the influence of atmospheric patterns as potential predictors as well as a prediction approach via time shifted MCA were performed. It is shown that SWE influences drought via downstream water/moisture transport from (high) mountain
regions. Especially SWE in the CR influences drought conditions in the Colorado River, Arkansas River, and Rio Grande catchments. Additionally, the Sierra Nevada and Appalachian Mountains show this relation on downstream regions to a lower extent. These findings of MCA were completely supported by PCA. Indeed, the maximizing of correlations in MCA has an added value compared to the use of PCA. The used drought indices and associated PET schemes are able to capture that relation although many important hydrological variables are not included in their calculation. The key for this lies in the process of
evapotranspiration which depends on energy and moisture availability. As the southern US are mainly arid and semiarid soil moisture is the limiting factor there. Thus, evaporation increases with a moisture increase provided by snowmelt. This results in lower sensible and higher latent heat with consequences for the temperature which enters in both PET schemes of the DIs.

Searching for predictors of SWE or drought, it is unavoidable to take atmospheric patterns into account. Especially ENSO has a clear impact on drought in the southern US and can be considered as a predictor there. Furthermore, SWE around
Colorado is influenced by ENSO, NAO, and PNA which can be predictors there, too. The time shifted MCA has shown the mainly affected months of drought on the basis of SWE in March. These are March to June/July for the already mentioned drainage basins.

There is still room for improvement and – besides subsection 4.3 – further investigation. A more regional focus solely on the western US might sharpen the findings and increase the SCFs as not all the differing climates of the US are put into
one analysis. However, further regionalization requires higher data resolution, whereas SPEI with .5° is a good start. In this context, the shortly introduced very high resoluted SNODAS data might contain a great potential for future work. Furthermore, a seasonal instead of annual investigation as well as not only SWE in March but differing months might improve the results and show more detailed links between SWE and drought, especially when considering atmospheric patterns. One might apply the used approach to other nival regions like Central Asia, Siberia, or Canada. In particular in Central Asia more differentiated
variables for snow and glaciers and their relation to drought may be investigated with the applied approach to find correlated spatio-temporal patterns of snow or glaciers and drought.

*Competing interests.*   The authors declare that they have no conflict of interest.

*Acknowledgements.*   We thank the European Centre for Medium Range Weather Forecast (ECMWF) for making the ERA–Interim data available, the National Center for Atmospheric Research (NCAR) for providing the sc-PDSI data, and the Consejo Superior de Investigaciones
Cientificas (CSIC) for providing the SPEI data. Furthermore, our thanks go to the Climatic Research Unit (CRU), NOAA / National Weather Service, and NOAA Earth System Research Laboratry (ESRL) for making the atmospheric indices available.





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
