# Peer review of "Influence of snow water equivalent on droughts and their prediction in the USA"

_Hydrology and Earth System Sciences, 2018_

## Referee Comment (RC1) · Anonymous Referee #1 · 5 Dec 2018

Summary

In this manuscript, Abel et al. examine potential connections between snow water equivalent and drought occurrence in the conterminous United States (CONUS). To this end, they conduct Maximum Covariance Analysis (MCA) – also known as Singular Value Decomposition – between monthly-averaged SWE values for March – obtained from the ERA-Interim reanalysis dataset (1979-2015) – and two drought indices (DIs): (i) the self-calibrating Palmer Drought Severity Index (sc-PDSI), and (ii) the Standardized Precipitation Evapotranspiration Index. The authors also perform Principal Component Analysis (PCA) on each field "to confirm or contradict the results from MCA". In a final stage, Abel et al. examine possible connections between standard climate indices, and SWE or DIs. MCA results show that connections between SWE and DIs

do exist, in particular between mountainous areas and downstream domains. Connections between ENSO and DIs are also displayed. Finally, the authors conduct time shifted MCA between march SWE and DIs to demonstrate the utility of the former for drought prediction.

The authors address a relevant problem for both scientific and operational communities. Further, the literature review is sound and the graphical results are nicely presented. Nevertheless, I found this manuscript extremely hard to follow and understand, mainly due to the lack of a coherent methodological flow. Therefore, this manuscript needs a major revision – including the proper use of English – before being considered for publication in HESS.

Major comments

1. Datasets used: In my opinion, using SWE datasets from a reanalysis sounds like a very odd approach, especially considering that the study domain is the US – where hundreds of SNOTEL sites have been operating in real-time for decades. The authors should, at the very least, include a robust evaluation of Era-Interim SWE across the CONUS, since the assessment they refer to (Brun et al. 2013) was conducted in Eurasia. Additionally, the authors should provide a justification of the atmospheric indices included in this study. Did they try including other standard indices? In my opinion, a better approach would be to directly explore interconnections between drought indices and spatial fields of variables such as sea surface temperature or geopotential height – extracted from Era-I. The authors can find many related examples aimed to predict hydrometeorological variables (e.g., Grantz et al. 2005; Block and Rajagopalan 2007; Mendoza et al. 2014; Ionita et al. 2015).

2. Approach: Although Singular Value Decomposition (SVD) has been widely used to understand the joint correlation structure between hydroclimatic fields (e.g., Rajagopalan et al. 2000; Sagarika et al. 2015), I'm not sure whether is the right approach to explore interconnections for prediction purposes. Did the authors consider using

[Figure]

Canonical Correlation Analysis (e.g., Salas et al. 2011) or Partial Least Squares Regression (e.g., Smoliak et al. 2010)? Also, I don't understand the rationale for including PCA in this study and highlighting those results in many sections (P8-L10; P10-L6). To the best of my understanding, PCA only looks for principal modes of variability in one field, while SVD looks for principal modes while maximizing covariance between two fields. I think the authors need to provide a better justification to include it (otherwise, delete those results from the manuscript). Finally, I think the authors should include, at the very least, a simple demonstration on how they would use their findings for drought prediction. This could be done, for example, by fitting linear regression models between PCs from SWE and DIs, and produce deterministic or ensemble forecasts using a cross-validation framework.

3. Conclusions: The authors state that "SWE influences drought via downstream water/moisture transport from (high) mountain regions". Did they really need to conduct all the analyses presented here to conclude this?

Minor comments

4. P1, L17-18: The first sentence in the introduction reads out of place. I suggest deleting. What do you mean with "high amounts of damage"?

5. I suggest the authors to carefully read Clark et al. (2001).

6. Section 2.1: The first paragraph reads out of place. I suggest deleting.

7. P3, L11: Do the authors mean in situ observations?

8. P3, L18: Reliable snowfall does not necessarily mean that snow depth and density estimates will be accurate.

9. Section 2.2: Is it possible to include sub-annual (i.e., seasonal) sc-PDSI values? How would the results change if 3-month averages of SPEI were used?

10. P5, L28: What do the authors mean with drying conditions? Less runoff?

[Figure]

11. P6, L10: This is the first and only time where "cross-validation" is mentioned. Did the authors actually conduct it?

12. All figures: Can you please add "a", "b", "c", etc. to the various panels? I think this would greatly improve the readability of this manuscript. Also, do you mean March SWE when referring to ERA-I in your figure titles and captions? If yes, I suggest to be more direct.

13. Figures 1-3: Can you please add a sub-panel with joint total variance as a function of mode?

14. P7, L3: I suggest the authors being more quantitative when reporting trends (e.g., adjust a linear regression, report confidence levels).

15. P9, L6: "Subsequently, the relation between SWE and sc–PDSI, respectively, and the two dominating patterns of the study area is investigated". I thought the authors were already doing this. Please re-word or delete.

16. Figure 7: It is really hard to distinguish correlations for annual, DJF and Oct-Mar without labels for each time scale. Would it be more logical to go from longer to shorter time periods? Also, are correlation values obtained from pulling together all time steps and points in the domain? Wouldn't it make more sense to plot correlation coefficients between leading modes from PCA (and not MCA) from each variable, and climate indices?

References

Block, P., and B. Rajagopalan, 2007: Interannual Variability and Ensemble Forecast of Upper Blue Nile Basin Kiremt Season Precipitation. J. Hydrometeorol., 8, 327–343, doi:10.1175/JHM580.1.

Clark, M. P., M. C. Serreze, and G. J. McCabe, 2001: Historical effects of El Nino and La Nina events on the seasonal evolution of the montane snowpack in the Columbia and Colorado River Basins. Water Resour. Res., 37, 741–757,

doi:10.1029/2000WR900305.

Grantz, K., B. Rajagopalan, M. Clark, and E. Zagona, 2005: A technique for incorporating large-scale climate information in basin-scale ensemble streamflow forecasts. Water Resour. Res., 41, W10410, doi:10.1029/2004WR003467.

Ionita, M., M. Dima, G. Lohmann, P. Scholz, and N. Rimbu, 2015: Predicting the June 2013 European Flooding Based on Precipitation, Soil Moisture, and Sea Level Pressure. J. Hydrometeorol., 16, 598–614, doi:10.1175/JHM-D-14-0156.1.

Mendoza, P. A., B. Rajagopalan, M. P. Clark, G. Cortés, and J. McPhee, 2014: A robust multimodel framework for ensemble seasonal hydroclimatic forecasts. Water Resour. Res., 50, 6030–6052, doi:10.1002/2014WR015426.

Rajagopalan, B., E. Cook, U. Lall, and B. K. Ray, 2000: Spatiotemporal variability of ENSO and SST teleconnections to summer drought over the United States during the twentieth century. J. Clim., 13, 4244–4255, doi:10.1175/1520-0442(2000)013<4244:SVOEAS>2.0.CO;2.

Sagarika, S., A. Kalra, and S. Ahmad, 2015: Pacific Ocean SST and Z 500 climate variability and western U.S. seasonal streamflow. Int. J. Climatol., n/a-n/a, doi:10.1002/joc.4442. http://doi.wiley.com/10.1002/joc.4442.

Salas, J. D., C. Fu, and B. Rajagopalan, 2011: Long-Range Forecasting of Colorado Streamflows Based on Hydrologic, Atmospheric, and Oceanic Data. J. Hydrol. Eng., 16, 508–520, doi:10.1061/(ASCE)HE.1943-5584.0000343. http://ascelibrary.org/doi/10.1061/%28ASCE%29HE.1943-5584.0000343.

Smoliak, B. V., J. M. Wallace, M. T. Stoelinga, and T. P. Mitchell, 2010: Application of partial least squares regression to the diagnosis of year-to-year variations in Pacific Northwest snowpack and Atlantic hurricanes. Geophys. Res. Lett., 37, n/a-n/a, doi:10.1029/2009GL041478. http://doi.wiley.com/10.1029/2009GL041478.

---

## Referee Comment (RC2) · Anonymous Referee #2 · 30 Jan 2019

The manuscript aims to evaluate the role of snow water equivalent (SWE) on drought in the United States and explore large scale predictors using drought metrics and re-analysis products via a type of principal component analysis.

The figures are beautiful and I like the detailed description of the maximum covariance analysis. The writing, however, is not of publication quality. The manuscript is often awkward and imprecise, and needs substantial revision. The references could be substantially improved, as significant efforts have been made towards the goal of the paper by numerous authors over the past decade; many of these efforts go un-cited in the paper. As an example, the ENSO discussion is completely out-of-date and needs to be improved.

The paper has numerous substantial scientific flaws and shortcomings that will need

to be addressed. I outline these below.

Some Major Problems:

1. 0.75° resolution ERA is not appropriate for examining snow in complex terrain without proving that it is against other datasets such as SNOTEL or a distributed model product. Like the other reviewer, I too am curious why the authors did not utilize the SNOTEL network. It is interesting to see the ERA-Interim snow product, however a comparison with SNOTEL is necessary and would add additional value to the analysis. I encourage this work as it would be helpful to know.

2. If drought is the question, extreme snow-free years cannot be excluded from the analysis (P3 L24). This represents a major flaw in the study and the analysis will need to be performed again with the inclusion of these years.

3. The Self-Calibrating Palmer Drought Severity Index is not appropriate for a drought analysis in a snow-dominated region, largely due to the lag effects of the simple bucket model. Though the results appear marginally reasonable, this is not a robust method as the sc-PDSI has a much longer memory (which the authors state) and was never designed for use in snow-covered environments. SPEI also does not address snow accumulation and melt. I recommend the authors attempt to develop a relevant metric for drought in snow-dominated environments, as this would be tremendously helpful and would really strengthen the contribution.

4. Using a set date or monthly average for SWE is not a useful indicator of SWE at the scales studied in the paper. I would recommend peak March SWE rather than a set date or average, but some locations may see peak SWE in April or possibly even early May. In other words, you will need to identify peak SWE timing at each gridpoint at a relevant scale, of which 0.75° is not sufficient due to the elevation differences of mountainous regions. Maybe try using the 6 km SWE estimates from the VIC model? While still not perfect, this is a great improvement over 0.75° resolution.
5. The monthly time scale of the analysis likely misses a lot of key precipitation, evapotranspiration, and temperature information. I recommend performing the analysis at a daily time scale since there is so much temperature and precipitation variability that could influence the results. If the results are consistent between daily and month timescales, this further strengthens your arguments.

6. It is clear that the authors are unfamiliar with the weather and climate of the regions they are studying. I recommend performing a detailed literature review of the key areas so that the statements made in the paper accurately reflect the hydroclimate of these regions.

7. The further investigations section should be worked into the paper as necessary and not be a standalone section.

8. No physical insight into how atmospheric patterns could be used to predict above or below average SWE was given. Are the patterns studied robust, do they vary, how might they change in the future? Much more analysis and detail is warranted as this is an important area of inquiry. We know ENSO has an impact on drought, but there is a lot of variability in the patterns of ENSO on cool season precipitation. The same goes for the other indices studied, yet the authors provide no insight into direct physical linkages or how these linkages may vary in time or space.

9. The alphabet soup of acronyms greatly detracts from the paper, please reduce the use of acronyms and make sure to proofread your paper prior to submission. Grammatical errors, poor spelling (e.g., Mexico does not have a 'k' in it), and poor sentence structure make the paper very difficult to review. If possible, please have a technical editor and English language editor review the paper prior to submission.

Summary: Because of these numerous methodological limitations and the poor writing quality, I do not believe the results are robust and I fail to understand how these results can inform improved decision making or seasonal forecasts of drought conditions in snow-dominated environments. Although the paper attempts to address relevant

questions within the scope of HESS, no novel concepts are provided nor are substantial conclusions reached, although with appropriate use of data and a significant improvement in the interpretation of results, I think the methodology could provide a substantial contribution. The references are also highly insufficient and could be expanded significantly. Unfortunately, at this point I do not recommend publication in HESS.

Some specific comments: P1 L1-2: These sentences are awkward, please revise.

P1 L1-7: Suggest revising this half of the abstract to be more precise.

P1 L8: I was looking forward to seeing this analysis in more detail, but did not see any explicit analysis of moisture transport in the paper. Suggest to remove this line if the paper does not evaluate moisture transport processes. That said, performing this analysis would benefit the paper overall.

P1 L9: Sentence starting with "Especially": What are you trying to say here, exactly?

P1 L10: What "link"? What is the relevance of higher ET and sensible heat fluxes? I am having a very hard time following the logic and connection of processes.

P1 L12: I don't recall skill being evaluated in a robust manner, were Heidke or Brier skill scores calculated?

P1 L13: The finding that ENSO is a good predictor for SWE in Colorado is inconsistent with nearly all the existing published literature. SWE is tightly controlled by precipitation, and there exists no significant relationship between precipitation and SWE in any part of Colorado at the climate division scale (see an example here: https://wrcc.dri.edu/Graphics/Plots/ENSO/soi_precip_co_div2.png?1)

P1 L17: This is a very strange way to open up a journal article. I am not saying it is wrong, but you might consider revising the introduction. This first sentence feels like a better opening sentence to the final paragraph in the introduction.

P1 L18: The title has "USA" but here you use "US". Either is fine, but please be consistent.

P1 L18: The sentence on droughts needs to be revised. There is a lot of good summary information here but few references and awkward phrasing.

\*\*\* At this point, I suggest completely revising the entire paper. I don't want to have to make the same comments on almost every sentence. While most sentences have some good and relevant information, many are in desperate need of revision to improve the clarity, style, and effectiveness of communicating the ideas. From here forward, I will try to only focus on the scientific content.

P2 L13: It appears you are trying to argue that snow, and its presence or absence, influences regional to global atmospheric circulation patterns on the same order as ENSO or the NAO. While snow certainly influences circulations (and has important implications for the initialization of numerical weather forecast models), this text oversells the role of snow and should be revised accordingly.

P2 L20: The ENSO discussion is far out of date and need massive revision to be modernized. I would also suggest to use the phrasing of "tends to lead to" rather than "leads to" since ENSO teleconnections are not perfectly stationary.

P2 L32: Please revise the final paragraph of the introduction. This is a critical component of your manuscript, and it feels like these questions are just thrown out there without being phrased in a careful, elegant manner.

P3 L7: Reanalysis products are not data. They can assimilate data, but they are not empirical data. I think use of reanalysis products are fine, but the spatial resolution of ERA-Interim ($0.75°$) is inappropriate for use in the highly complex terrain of the western US. You will need to supplement this analysis with actual data (e.g., SNOTEL network) or perhaps with other model products (e.g., SNODAS). If you are able to show that $0.75°$ horizontal resolution reanalysis output does a reasonable job, that is excellent

information. But first you must convince us that it is! This is a mandatory step in the analysis.

P3 L20: No, the amount of SWE at the beginning of the melting period is absolutely NOT the only relevant amount. I strongly recommend you think deeply about why this is not the case before you continue this work. Doing so will open up a much better understanding about the problem you are approaching, and you might discover a more novel way to address the problem than currently exists! Furthermore, April 1 is now known to NOT represent the best date (see for example Margulis et al. 2016 J. Hydromet).

P3 L21: I do not think the monthly SWE product is appropriate for this study; it may miss key aspects such as late season storm events or warm spells. Again, doing a detailed comparison with a daily dataset like SNOTEL would help demonstrate if this is (or is not) the case, and would help ensure robust results from your study.

P3 L23: PLEASE DO NOT EXCLUDE EXTREME YEARS! If the purpose of the paper is to study drought in snow-dominated systems, these are some of the most important years to study!

P5 L25: Please, never start your results section with results that are not shown! Provide readers with these results, or at the very least, add them as supplementary material.

P5 L27-29: Good place to provide actual references to the literature here.

P6 Figure: What is t on the right hand x-y plot and what do the values on the abscissa represent?

P6 L4: Is this correlation significant? If so, at what level?

P6 L8: I may be missing something, but if the regression results are not shown, how can we be sure that the correlation maps are "capturing the dominating trends well"?

P7 L9: What exactly are you trying to say here? Please clarify this sentence.

P7 L12: Again, please provide statistical significance for r values.

P10 L1: This paragraph could be written much more clearly.

P10 L14: Negative ones? Do you mean negative correlations?

P10 L15-16: This sentence is very confusing, please rewrite.

P11 L13: I suggest performing this regional analysis with a higher resolution snow dataset. I think this would strengthen the paper.

P13 L6: Do you mean these areas are the headwaters of important rivers?

P13 L10: I don't see how SWE and sc-PDSI, which are shown spatially at 0.75° resolution, would be connected through rivers without an analysis of streamflow. SWE and streamflow or sc-PDSI and streamflow should be connected, is that what you mean?
* * *